# Disappearing Colorectal Liver Metastases: Do We Really Need a Ghostbuster?

**DOI:** 10.3390/healthcare10101898

**Published:** 2022-09-28

**Authors:** Alessandro Anselmo, Chiara Cascone, Leandro Siragusa, Bruno Sensi, Marco Materazzo, Camilla Riccetti, Giulia Bacchiocchi, Benedetto Ielpo, Edoardo Rosso, Giuseppe Tisone

**Affiliations:** 1Department of Surgical Science, University of Rome “Tor Vergata”, 00133 Roma, Italy; 2Department of Surgery, University Campus Bio-Medico di Roma, 00128 Roma, Italy; 3Hepatobiliary and Pancreatic Surgery Unit, Hospital del Mar. Universitat Pompeu Fabra Barcelona, 08003 Barcelona, Spain; 4Unité des Maladies de l’Appareil Digestif et Endocrine, Centre Hospitalier de Luxembourg, 1210 Luxembourg, Luxembourg

**Keywords:** colorectal neoplasms, liver neoplasms, magnetic resonance imaging, disappearing colorectal liver metastases, colorectal cancer, hepatobiliary surgery, systemic therapy, ultrasonography, hepatectomy, prognosis

## Abstract

The development of new systemic treatment strategies has resulted in a significant increase in the response rates of colorectal liver metastases (CRLM) in the last few years. Although the radiological response is a favorable prognostic factor, complete shrinkage of CRLM, known as disappearing liver metastases (DLM), presents a therapeutic dilemma, and proper management is still debated in the literature. In fact, DLM is not necessarily equal to cure, and when resected, pathological examination reveals in more than 80% of patients a variable percentage of the tumor as residual disease or early recurrence in situ. Moreover, while a higher incidence of intrahepatic recurrence is documented in small series when surgery is avoided, its clinical significance for long-term OS is still under investigation. In light of this, a multidisciplinary approach and, in particular, radiologists’ role is needed to assist the surgeon in the management of DLM, thanks to emerging technology and strategy. Therefore, the aim of this review is to provide an overview of the DLM phenomenon and current management.

## 1. Introduction

Colorectal cancer (CRC) is the third leading cause of cancer-related death worldwide in both males and females [1]. Over 15% of patients present liver metastases at the time of diagnosis, and up to 50% develop them within 3 years [2]. Among patients with metastatic colorectal cancer (mCRC), approximately 20% of them survive beyond 5 years from diagnosis [3].

Although systemic chemotherapy is broadening its indications in patients with mCRC, surgical treatment is the only definitive treatment, and thus, multidisciplinary decision-making is mandatory in the different subsets of patients (e.g., multiple sites of liver-limited disease that is not resectable, or resectable disease with a high risk of recurrence). In this subset of patients, multimodal ablative therapy may result in improved outcomes compared with continued systemic therapy alone [4,5,6]. Currently, the number of patients with colorectal liver metastases (CRLM) who are a candidate for hepatic resection is significantly increased due to new surgical strategies and oncological therapies, which increase the rate of resectable disease [7,8,9].

In fact, while upfront surgery is the treatment of choice in patients with resectable disease, over 60% of patients with CRLM present an objective response to chemotherapy, according to RECIST 1.1. criteria, and this is usually an indicator of favorable prognosis and can be resected after chemotherapy [10,11]. Moreover, a complete radiological response with the disappearance of CRLM can occur in a minority of patients. This phenomenon, named disappearing liver metastases (DLM), also known as vanishing, missing or ghost CRLM, is based on the complete shrinkage of liver metastases after chemotherapy. DLM incidence is highly variable, depending on the accuracy and quality of restaging imaging with up to 37% of patients undergoing systemic treatments [12].

Among different strategies, magnetic resonance imaging (MRI) and contrast-enhanced intraoperative ultrasound (CEIOUS) are the most accurate imaging modalities and should be adopted routinely for detecting DLM [13]. However, the optimal management of DLM is still controversial due to the uncertainty of residual microscopic disease and effective long-term outcomes in resected versus un-resected patients [14,15]. This review aims to provide an overview of the state of the art in the management of DLM, which is still a critical challenge in clinical practice.

## 2. Systemic Treatment of CRLM

The liver is the most common metastatic site for CRC patients, and 20 to 34% present with synchronous hepatic involvement at diagnosis [16,17,18,19]. Chemotherapy represents the best treatment for most of these patients. However, especially in the case of metastatic liver-limited disease, a conversion approach could be considered, and surgical re-evaluation could be planned every 2 months from the beginning of therapy.

To date, there is no standard for this attempt, and chemotherapy backbone choice, represented by 5-fluorauracil/leucovorin (5FU/LV), irinotecan (IRI) and oxaliplatin (OXA), depends on clinical preference and patient clinical conditions. Previous retrospective studies on initially unresectable mCRC treated with 5FU/LV combined with irinotecan (FOLFIRI) or oxaliplatin (FOLFOX) showed positive conversion rates for both the combined schemes (32.5 and 40%, respectively) [20,21]. However, secondary hepatotoxicity due to both irinotecan and oxaliplatin, represented by steatohepatitis and sinusoidal liver injury, respectively, does not support the use of one drug or the other [22,23].

Additionally, in 2007, an alternative treatment was presented for ECOG performance status 0 patients: the triple combination of 5FU/LV, oxaliplatin and irinotecan (FOLFOXIRI). FOLFOXIRI, which resulted in a paradigmatic shift. Despite the increased incidence of high-grade (G3/4) adverse events, FOLFOXIRI demonstrated a superior response rate (66% vs. 41%), prolonging progression-free and overall survival (OS) when compared with FOLFIRI [24].

Besides FOLFOXIRI, prior therapy international guidelines suggest that all patients with mCRC should be tested for exons 2, 3 and 4 of rat sarcoma virus (RAS) (K- and N-) and BRAF genes mutations as well as for microsatellite instability (MSI) proteins or mismatch repair (MMR) gene deficiencies. These genetic tests are needed to determine a tumor’s biological features and support clinicians in selecting the best monoclonal antibody to add to chemotherapy [25].

RAS/RAF wild-type patients are eligible to be treated with anti-epidermal growth factor (EGFR) monoclonal antibodies (e.g., cetuximab and panitumumab). Previous posthoc analysis of patients’ liver conversion rate treated with FOLFOX or FOLFIRI alone or plus cetuximab showed a positive trend from the addition of anti-EGFR in KRAS wild-type selected cohort (60 vs. 32%) as for the CELIM trial [7]. Similar results were obtained in a single-centre pan-Asian experience, in which KRAS wild-type liver-only mCRC were randomized to receive chemotherapy alone or with cetuximab, confirming the superiority of the combined treatment on FOLFOX or FOLFIRI alone in terms of R0 and objective response rate (25.7% vs. 7.4% and 57.1% vs. 29.4%, respectively) [26].

Considering the triple drug combination efficacy, the phase II trial VOLFI tested the efficacy of panitumumab when combined with FOLFOXIRI vs. chemotherapy alone. Overcoming the predefined objective response rate cut-off of 75% (87.3 vs. 60.6%), data support FOLOFOXIRI plus panitumumab [27]. Finally, further evidence will be available from the Italian phase III TRIPLETE trial. The TRIPLETE trial, which aimed to compare FOLFOXIRI plus panitumumab vs. standard of care (FOLFOX Panitumumab), just finished enrolment, and their results will soon be published [28].

The presence of RAS/RAF mutations contraindicates anti-EGFR agent use due to intrinsic drug resistance [29], while promoting the use of the anti-vascular endothelial growth factor (VEGF) monoclonal antibody bevacizumab in the first line setting. However, data about the additive effect of this agent on two-agent chemotherapy in terms of resection rate are controversial. While negative results were obtained in a big cohort of 1400 mCRC patients, on the other hand, the BECOME trial results sustained the addition of monoclonal antibodies to the oxaliplatin-based scheme [30].

In the BECOME trial, among 241 initially un-resectable RAS mutant patients, randomized to receive FOLFOX with bevacizumab or alone, 27 of 121 in the experimental arm (vs. 7 of 120 patients of the chemo-only arm) underwent R0 surgery and reached a statistically significant objective response rate (*p* < 0.01). Stronger-performing results on overall response rate, as above, were obtained in several phase II and III trials by the triplet scheme FOLFOXIRI with bevacizumab [31,32,33]. In this regard, in a pooled analysis of 3 prospective Italian trials on more than 200 patients with liver-limited disease receiving the triple combination plus anti-VEGF, 69% of patients reached a radical surgery attempt, resulting in R0 for 30.7% of them. Moreover, R0/1 resected patients had longer survival compared to other patients, both in terms of PFS and OS, independently from mutational status, including BRAF mutant tumours, which still represent an oncological challenge due to their aggressive biology [34].

The advent of immunotherapy changed the treatment paradigm in MSI-high or MMR-deficient (dMMR) mCRC patients. Anti-programmed death (anti-PD1) antibody, alone or in combination with anti-CTLA4, showed significantly longer survival periods, reaching higher rates of complete and sustained complete response in this cohort compared to previously obtained chemotherapy results. The overall response rate of pembrolizumab alone, nivolumab alone or in combination with ipilimumab was 43.8, 31.1 and 55%, respectively [35,36,37]. In light of this, future studies will need to investigate the potential role of surgery or local therapy in this subset of patients after a first immune checkpoint inhibitor (ICI) approach.

## 3. Predictors of Complete Radiological Response after Chemotherapy

In the last decades, several factors have been identified as possible predictors of developing DLM. Among them, low carcinoembryonic antigen (CEA) levels at diagnosis and its normalization after chemotherapy, patients younger than 60 years, re-staging with magnetic resonance image (MRI), synchronous disease, size and number of LM at diagnosis, number of chemotherapy cycles, agents included in regimens and the use of hepatic arterial infusion (HAI) chemotherapy have been associated with the confirm of DLM with a complete pathologic response [38,39,40,41,42].

Currently, the focus is shifting to tumor burden. Xu et al. showed that the early primary T stage (T1–2 vs. T3–4, OR [95% CI]: 3.131 [1.213–8.082], *p* = 0.018) was an independent predictor of pathologic complete response after preoperative chemotherapy [43]. Factors predisposing to the development of DLM are summarized in Table 1.

However, there is no absolute consensus on the criteria for complete pathologic response, and in fact, several authors analyzing the correlation between the size of CRLM prior to chemotherapy and the onset of DLM revealed wide variability in the mean or median size of CRLM. Moreover, most of these studies were performed before the advent of target therapy.

The likelihood of developing a “true” complete pathological response after chemotherapy is higher in patients with a combination of these factors, and it is associated with both prolonged survival and decreased risk of recurrence [42].

## 4. The “Histological Truth” behind Complete Radiological Response

According to the new guidelines established through RECIST 1.1 criteria [10], the goal of the treatment of solid tumors is defined as the disappearance of the target lesions. However, a trend toward an increased rate of DLM leads the surgeon to a decision-making dilemma: to resect or not the sites of the now-missing targets. Data collected through an international survey [40] revealed that hepatobiliary surgeons do not have a unified attitude in the management of this challenging setting. In several series, the pathologic examination of the DLM on imaging revealed a variable percentage of macroscopic or microscopic residual disease or early recurrence in situ up to more than 80% [12,44]. The histological results obtained from the analysis of the resected DLM are shown in Table 2.

## 5. The Role of the Radiologist

It is sometimes difficult in routine radiological practice to detect DLM. Most of the time, when patients have MRIs to monitor their disease, they have already received neoadjuvant therapy. The clinician then asks about the status of the liver parenchyma and the evolution of liver metastases, which are difficult to detect on ultrasound and/or MRI due to the effect of primary medical treatment [45].

In order to detect residual tumors, a classical protocol (T1 without and with multiphasic gadolinium injection, a diffusion sequence and with ADC mapping, a T2 Fat-SAT sequence) is often used. In most cases, these sequences are sufficient to assess CRLM but are not the only ones used.

This is when the role of the radiologist is pivotal to detect any sign of remnants, which are not always visible on other sequences [46].

For instance, some metastatic lesions are no longer visible on T1 sequences without and with injection and are, therefore, extremely difficult to visualize. Sometimes the only sequence that allows the radiologist to detect residual tumor lesions is the diffusion-weighted sequence. Indeed, at high “B” values, these “vanishing lesions”, which are not visible on the other sequences, may still show a diffusion restriction. We, therefore, know that they are the sites of residual tumor cells [47].

However, when the classical protocol is not sufficient to assess CRLM, other strategies are needed. In the past two decades, MRI has demonstrated enhanced accuracy thanks to novel hepatocyte-specific contrast agents such as gadolinium ethoxybenzyl dimeglumine (Gd-EOB-DTPA) [55]. Gd-EOB-DTPA, which behaves similarly to traditional contrast agents in the dynamic phases, adds substantial information in the hepatobiliary phase, detecting and characterizing focal liver lesions or diffuse liver disease [56]. The importance of Gd-EOB-DTPA has been described in a retrospective analysis by Morin et al., where among 110 patients with liver metastasis, in 43% of patients, Gd-EOB-DTPA revealed a different number of liver lesions, and it potentially modified surgical planning in more than 17% of patients [55]. Furthermore, as the future of radiology and hepatobiliary surgery will tell us, the role of the radiologist is changing, and a radiologist must be able to use artificial intelligence tools, including 3D reconstruction and radiomics, that surely will be involved in solving the DLM dilemma [57,58,59].

In fact, radiologists’ role is also to help the surgeon plan the surgical procedure by positioning the residual lesion (sometimes visible only on the diffusion-weighted sequence) in relation to the other vessels using 3D modelling. In this way, the surgeon in the operating room will be able to locate the lesion more easily during the operation, because, very often, even with intra-operative ultrasound, the surgeon cannot find the residual tumor site [60]. Moreover, 3D models can be integrated into navigated image guidance systems (IGS) to guide surgery and provide additional information to the surgeon, merging all the information provided from the CEIOUS and preoperative imaging [61].

The surgeon will therefore have to trust the 3D model previously designated by the radiologist on the basis of the MRI and the diffusion sequence. Furthermore, 3D printed models generated from medical images that are graspable and patient-anatomy-specific will improve the understanding of preoperative surgical liver anatomy and, eventually, surgical resection accuracy [62].

However, radiologists’ role is not focused on guiding surgical plans and positioning the residual lesions, but their role is critical in multidisciplinary teams. For instance, in the initial setting, during a multidisciplinary assessment, radiologists should detect the DLM high-risk lesions, for which a fiducial marker could reduce the risk of DLM, representing the real ghostbuster [63,64]. Image interpretation by radiologists could guide initial therapy discussions as well as interpret post-treatment imaging following liver-directed therapy, providing additional information to all the members of multidisciplinary team. Moreover, radiologists in the perioperative settings could help the surgeon beyond CEIOUS, with local percutaneous treatment or in case of complications (e.g., stenting or abdominal drainage) [65,66,67].

## 6. Novel Strategies in the Management of Patients with DLM

Although radiological morphology could be considered a surrogate of complete pathologic response (CPR), the finding of DLM in cross-sectional imaging does not mean that those lesions have been cured. Furthermore, a recent consensus on hepatic resection for CRLM established that surgery should include all sites of liver metastases described before chemotherapy [68]. However, no recommendation based on hard evidence supports whether DLM should be resected or left in situ [13].

In fact, surgical treatment is the only curative treatment for CRLM, and the development of novel chemotherapy, plus the advance in radiology, as mentioned before, and surgery has ameliorated the clinical outcome of CRLM [69].

Advocates of DLM resection mention the low incidence of CPR (20%) and higher rates of CRLM recurrence in situ (70%) due to the microscopic residual disease or the presence of a supportive microenvironment for tumor relapse [19,70,71,72]. In surgical planning, even with the 3D reconstruction, CEIOUS represents a valuable instrument in the hand of the surgeon that can decipher the DLM dilemma, increasing the detection rate of DLM when compared with Gd-EOB-DTPA alone. In fact, Gd-EOB-DTPA + CEIOUS, with a sensitivity of 93% and specificity of 73%, could detect the clinically relevant DLM with viable tumors, thus at risk of local recurrence [49].

Regarding the surgical procedure, one-stage parenchymal-sparing hepatectomy (PSH), when feasible, had comparable safety and efficacy when compared with anatomical resection (AR) and did not compromise oncological outcome, reducing the hepatic parenchyma resected [4]. Furthermore, whereas R1 resections increase the likelihood of locoregional recurrence and redoing liver surgery, R1 resection with detachment from major intra-hepatic vessels (R1Vasc) was demonstrated to achieve an outcome equivalent to R0 resection [4,73].

However, in some cases, complete eradication of the original site of the disease could be very complex to perform due to the localization deep in the parenchyma; moreover, an extensive resection could involve the risk of an insufficient liver remnant. For these reasons, traditional surgery has moved towards new strategies such as two-stage hepatectomy, portal vein embolization (PVE) and associating liver partition and portal vein ligation for staged hepatectomy (ALPPS) [74].

Finally, alternatives to surgical strategy that should always be evaluated are cryotherapy and radiofrequency ablation (RFA). Microwave ablation (MWA) can also be evaluated in the case of extrahepatic disease, the deepest position where the surgical procedure could compromise too much parenchyma, patients’ comorbidities [75]. For instance, a recent metanalysis indicates how MWA can represent a valid tool to treat CRLM, especially in those smaller than 3 cm, alone or in combination with hepatectomy to expand the pool of resectable patients [65,75].

Besides, a surgical strategy should be driven not only by the anatomical and technical variables but also considering molecular disease characteristics, such as RAS mutational status, to decide margin width and, in the case of DLM, the indication to proceed with resection [76,77,78].

Some studies have already started to compare different treatment approaches (watchful waiting vs. resection). Goere et al. reported a 5-year OS and recurrence-free survival (RFS) of 80% and 23%, respectively and a median recurrence time between 13.8 and 21 months. In their study cohort, adjuvant chemotherapy with oxaliplatin-based hepatic arterial infusion resulted in a lower rate of intrahepatic relapse [50]. Similar results were confirmed by Tanaka et al., who reported, after a median follow-up of 44 months, a lower recurrence rate in DLM resected patients when compared with watchful waiting (24.4% vs. 40.7%, respectively) [41].

Another study by Owen et al. demonstrated no difference in terms of RFS between DLM left in situ (360 days) and resected (483 days) [46].

However, the clinical significance of intrahepatic recurrence may not automatically determine a detrimental effect of OS. In fact, although Van Vledder et al. reported a higher rate of 3 years intrahepatic recurrence in the surveillance group when compared to the surgical group, in the same series, no survivorship advantage was reported at 1, 3, and 5 years [42]. In light of this, current evidence demonstrates that DLM resection could determine a benefit in terms of RFS, and DLM resection seems not to affect survivorship despite the high risk of residual disease in the previous tumor bed [41,42].

Moreover, the COVID-19 emergency determined a change in colorectal patients, leading to a higher rate of delayed presentation, delayed treatment, and oncological emergency with novel real-life evidence that could determine a change in clinical practice [79,80,81,82,83].

On the basis that maximizing resection of CRLM still remains the main objective, some authors have experimented with marking lesions at high risk of disappearing with a fiducial prior to initiation of chemotherapy or immediately before surgery. Vujic et al. demonstrated the importance of positioning a CT-guided marker in DLM after neoadjuvant treatments and observed a complete histological response in only 18% of resected sites [63,64].

However, the fiducial marker placement is not without possible complications; therefore, it should be considered, especially in CRLM at high risk of being missed. Kepenekian et al. described the fiducial placement in lesions smaller than 25 mm in diameter, deeper than 10 mm in the hepatic parenchyma and sited outside the field of a planned resection [63].

Fiducial placement could change the paradigm in the high-risk DLM treatment by allowing the surgeon to move towards tailored and more radical resections.

## 7. Conclusions

The complete curing of CRLM with systemic therapy is a rare phenomenon that now occurs in less than 5% of cases. However, thanks to the development of innovative oncological strategies, it is safe to state that in the future, a higher rate of patients will develop DLM [84].

Still, most complete radiological responses actually harbor macro or microscopic foci of residual disease. Yet, resection of DLM can be technically troublesome. For this reason, it is necessary to perform detailed restaging after and during chemotherapy with accurate localization of all sites of CRLM previously described as the key point for the correct treatment [85]. An algorithm for preoperative DLM management has been proposed (Figure 1).

Despite preliminary experiences demonstrating how watchful waiting could represent a safe alternative to DLM, future studies are needed to determine the risk of relapse in DLM. While waiting for artificial intelligence, radiomics or other innovative diagnostic techniques, fiducial marker placement in CRLM at high risk to become DLM will represent, in the next years, the practice-changing procedure that will help as a ghostbuster, reducing the rate of undetected DLM and providing a guide to solve the DLM dilemma with resection of the fiducial tissue with adequate margins.

## Figures and Tables

**Figure 1 healthcare-10-01898-f001:**
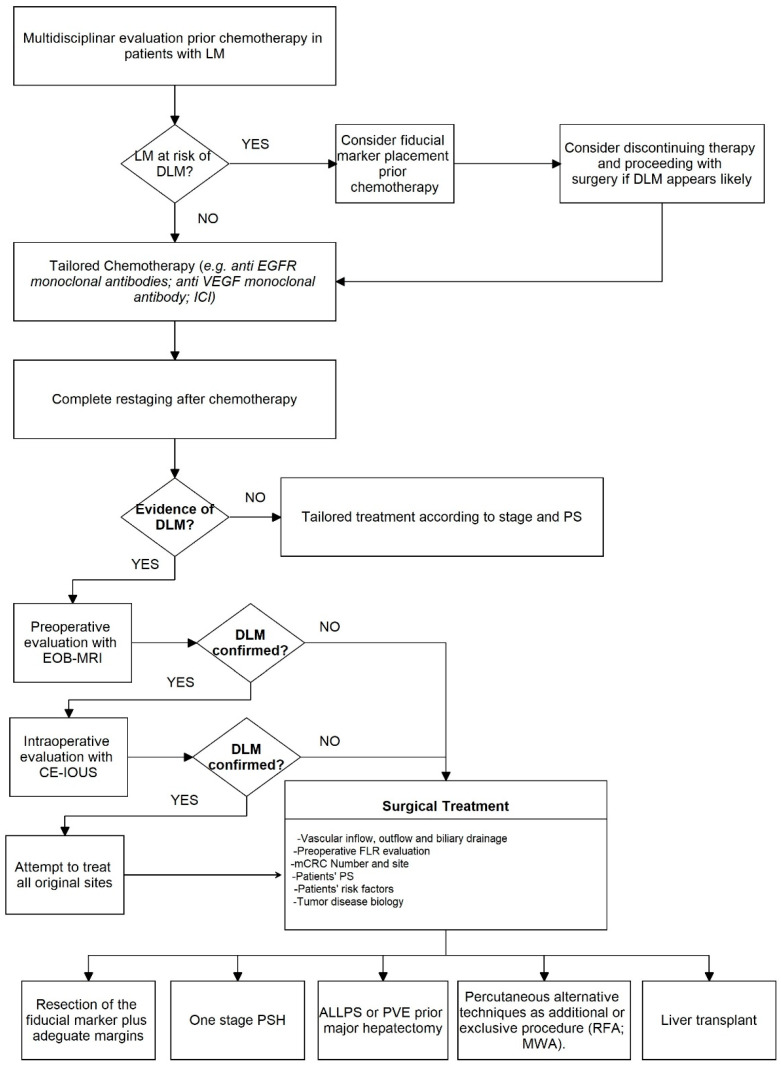
Preoperative surgical algorithm for DLM. LM: liver metastasis; DLM: disappearing liver metastasis; EGFR: epidermal growth factor receptor; VEGF: vascular endothelial growth factor; ICI: immune checkpoint inhibitor; EOB-MRI: gadoxetic acid-enhanced magnetic resonance imaging; CE-IOUS: contrast-enhanced intraoperative ultrasound; FLR: future liver remnant; PS: performance status; PSH: parenchymal-sparing hepatectomy; ALLPS: associating liver partition and portal vein ligation for staged hepatectomy; PVE: portal vein embolization; RFA: tadiofrequency ablation; MWA: microwave ablation; mCRC: metastatic colorectal cancer.

**Table 1 healthcare-10-01898-t001:** Predictive factors of DLM and complete response. LM: liver metastasis, DLM: disappearing liver metastasis, RR: risk ratio; CEA: carcinoembryonic antigen; HAI: Hepatic arterial infusion; OR: Odd ratio.

Author (Year)	Predictors
**Benoist (2006)** [44]	Mean maximum size of LM at diagnosis (cm) (DLM: 2.2 ± 1.5 vs. no DLM: > 4.5)
**Adam (2008)** [39]	Age ≤ 60 years (RR = 4.1; *p* = 0.03) Size of LM at diagnosis ≤ 3 cm (RR = 3.1; *p* = 0.05) CEA level at diagnosis ≤ 30 ng/mL (RR = 5.6; *p* = 0.03)
**Tanaka (2009)** [41]	Smaller size at diagnosis (mm) (DLM: 15.9 ± 14.3 vs. no DLM: 24.4 ± 22.3; *p* < 0.001) Fewer microscopic cancer deposits surrounding macroscopic tumors (%) (DLM: 21.7 vs. no DLM: 52.5%; *p* < 0.05)
**Auer (2010)** [38]	HAI chemotherapy (OR 6.2; *p* = 0.02) Inability to observe LM on MRI (OR 4.7; *p* = 0.005) Normalization of CEA levels (OR 4.6; *p* = 0.006)
**van Vledder (2010)** [42]	Smaller size of LM (cm) (DLM: 1.0 (0.3–3.5) vs. no DLM: 2.1 (0.4–16); *p* < 0.001) No. of cycles of preoperative chemotherapy (OR 1.18; *p* = 0.03) No. of LM at diagnosis >3 (OR 13.1; *p* < 0.001)
**Ferrero (2012)** [45]	No. of cycles of preoperative chemotherapy (OR 0.231; *p* = 0.022)
**Owen (2015)** [46]	Synchronous LM (OR 11.25; *p* = 0.015) No. of LM at diagnosis (DLM: 14.5 (4–39) vs. no DLM: 3.5 (1–30); *p* < 0.001)
**Kim (2016)** [40]	Mean size of LM at diagnosis (mm) (DLM: 6.8 ± 3.4 vs. no DLM: 9.33 ± 4.1; *p* < 0.001)
**Park (2017)** [47]	No. of LM at diagnosis (DLM: 6.0 ± 2.5 vs. no DLM: 4.1 ± 2.6; OR 1.390; *p* = 0.001)
**Tani (2018)** [48]	No. of LM [DLM: 14.5 (4–39) vs. no DLM: 3.5 (1–30); *p* < 0.0001] Smaller size of LM (cm) (DLM: 0.6 (0.4–2.0) vs. no DLM: 1.4 (0.3–13.0); *p* < 0.0001) Oxaliplatin-based chemotherapy (%) (DLM: 100% vs. no DLM: 75.8%; *p* = 0.017)
**Oba (2018)** [49]	Median size of LM at diagnosis: 8 mm (range: 3–34 mm)
**Xu (2020)** [43]	Size of LM < 3 vs. >3 cm (OR: 20.542; *p* = 0.003) Preoperative CEA levels ≤ 20 vs. >20 ng/mL (OR: 7.656; *p* = 0.049) Primary T stage T1–2 vs. T3–4 (OR: 3.131; *p* = 0.018) Primary tumor location (right vs. left-sided) (OR: 2.808; *p* = 0.017)

**Table 2 healthcare-10-01898-t002:** The histological results obtained from the study of the resected DLM. DLM: disappearing liver metastasis; LM: liver metastasis.

Author (Year)	No. of Patients	No. of Patients with One or More DLMs (%)	Residual Disease after RESECTION (%)
Elias (2004) [50]	104	15 (14.4%)	6/11 patients (54.5%)
Benoist (2006) [44]	586	38 (7%)	12/15 LM (80%)
Elias (2007) [51]	228	16 (7%)	8/16 patients (50%)
Adam (2008) [39]	767	n.s.	2/2 patients (100%)
Tanaka (2009) [41]	63	23 (36.5%)	0/28 LM (0%)
Van Vledder (2010) [42]	168	40 (23.8%)	41/67 LM (61.2%)
Auer (2010) [38]	435	39 (8.9%)	24/68 LM (35.3%)
Ferrero (2012) [45]	171	33 (19.3%)	33/45 LM (73.3%)
Ono (2012) [52]	125	n.s.	0/2 LM (0%)
Arita (2014) [53]	72	11 (15.3%)	16/25 LM (64%)
Sturesson (2015) [54]	179	29	32/36 LM (88.9%)
Owen (2016) [46]	23	11 (47.8%)	21/36 patients (58.3%)
Oba (2018) [49]	185	59 (32%)	3/68 patients (4%)

## Data Availability

Not applicable.

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
