# Peer review of "Disappearing Colorectal Liver Metastases: Do We Really Need a Ghostbuster?"

_healthcare, 2022, doi:10.3390/healthcare10101898_

Round 1

Reviewer 1 Report

This is a well written paper, analyzing a challenging topic as the disappearing colorectal liver metastases (CLMs). Authors narratively reviewed the literature and correctly reported the multispecialistic approach about CLMs.

The description of systemic treatments is accurate as well as the predictors of complete radiological response after chemotherapy. The role of radiologist is highlighted.

However, the paper is still missing some important points about this complex scenario.

The title focuses on the role of a ghostbuster and the abstract cites the dilemma of a “proper surgical management” but only a small part of the article describes the role of surgeon in this “surgical problem”.

In the introduction, systemic therapy is defined the cornerstone of the treatment of CLMs but surgery is the only curative treatment. The majority of metastatic disease could be tackled by parenchymal-sparing upfront surgery with no conversion chemotherapy needed.

The analysis on systemic treatment for CLMs is accurate and complete as well as the description of predictors of complete radiological response.

Looking at radiologist’s role some important point must be clarified: first of all, the importance of preoperative imaging, specifically focused on GD-EOB-MRI and not the simple use of gadolinium. Moreover, the radiologist has not only a role “per se” but also in a multidisciplinary setting, guiding other specialists on the correct analysis of imaging, this concept should be stressed.

The role of 3D reconstruction is predominant, specifically focusing on DLM, with the possibility to intraoperatively find disappearing foci with the fusion of images between pre-op MRI, 3D reconstruction and intraoperative ultrasound (IOUS).

Finally, the role of the surgeon. The title cites the ghostbuster but only few words are spent on the most important point of the discussion. What to do with a DLM? Which is the sensitivity of intraoperative ultrasound? Is there still an advantage in using it in the era of new EOB-MRI) Which are the advantages of the CE-IOUS? Kokudo et al. in 2017 wrote a cornerstone on this topic that must be taken into account. Sometimes, a DLM is just a very small, almost undetectable lesion until a contrast-enhanced intraoperative ultrasound is performed. It should be distinguished the real DLM from an almost undetectable shrinkage lesion.

About surgical techniques per se: it has been cited about ALPSS, two-stage, portal vein embolization. In this scenario an important role has to be recognized to parenchymal-sparing one stage hepatectomies and speaking about margin width, the possibility for CLMs of R1vasc resection. Few words should be spent on the role of ablation for deepest foci, where a surgical resection may compromise too much parenchyma.

Some spell check required:

Row 129: ICI -> specify immune checkpoint inhibitor

Row 131: factors have been IDENTIFIED

Row 138: the focus is shifting ON tumor burden

Overall, the paper deals with a challenging aspect that does not find in literature a clear and a unique view. For this reason, an overview like this deserves to be published, but major revision of existing literature and a more comprehensive overview, even in its narrative review setting, must be done.

Author Response

We would like to thank the Editors and Reviewer for reviewing and give us the opportunity to improve our manuscript
“Disappearing colorectal liver metastases: do we really need a ghostbuster?”

The Reviewer raised a number of important comments about our work, in view of which we have revised our
manuscript to address all the issues. We believe that this has now substantially improved the manuscript, ameliorating
its impact and clarity.
Please find attached a point-by-point response of the issues raised and our revised manuscript.
We thank you for reconsidering our paper for publication in Healthcare.
Reviewer #1,
This is a well written paper, analyzing a challenging topic as the disappearing colorectal liver metastases (CLMs).
Authors narratively reviewed the literature and correctly reported the multispecialistic approach about CLMs. The
description of systemic treatments is accurate as well as the predictors of complete radiological response after
chemotherapy. The role of radiologist is highlighted. However, the paper is still missing some important points about
this complex scenario.
The title focuses on the role of a ghostbuster and the abstract cites the dilemma of a “proper surgical management” but
only a small part of the article describes the role of surgeon in this “surgical problem”. 
Overall, the paper deals with a challenging aspect that does not find in literature a clear and a unique view. For this
reason, an overview like this deserves to be published, but major revision of existing literature and a more
comprehensive overview, even in its narrative review setting, must be done.
In the introduction, systemic therapy is defined the cornerstone of the treatment of CLMs but surgery is the only
curative treatment. The majority of metastatic disease could be tackled by parenchymal-sparing upfront surgery with no
conversion chemotherapy needed. 
The analysis on systemic treatment for CLMs is accurate and complete as well as the description of predictors of
complete radiological response.
Looking at radiologist’s role some important point must be clarified: first of all, the importance of preoperative
imaging, specifically focused on GD-EOB-MRI and not the simple use of gadolinium. Moreover, the radiologist has not
only a role “per se” but also in a multidisciplinary setting, guiding other specialists on the correct analysis of imaging,
this concept should be stressed.
The role of 3D reconstruction is predominant, specifically focusing on DLM, with the possibility to intraoperatively
find disappearing foci with the fusion of images between pre-op MRI, 3D reconstruction and intraoperative ultrasound
(IOUS).

Finally, the role of the surgeon. The title cites the ghostbuster but only few words are spent on the most important point of the discussion. What to do with a DLM? Which is the sensitivity of intraoperative ultrasound? Is there still an
advantage in using it in the era of new EOB-MRI) Which are the advantages of the CE-IOUS? Kokudo et al. in 2017 wrote a cornerstone on this topic that must be taken into account. Sometimes, a DLM is just a very small, almost undetectable lesion until a contrast-enhanced intraoperative ultrasound is performed. It should be distinguished the real DLM from an almost undetectable shrinkage lesion. 
About surgical techniques per se: it has been cited about ALPSS, two-stage, portal vein embolization. In this scenario an important role has to be recognized to parenchymal-sparing one stage hepatectomies and speaking about margin
width, the possibility for CLMs of R1vasc resection. Few words should be spent on the role of ablation for deepest foci, where a surgical resection may compromise too much parenchyma.
 Some spell check required: Row 129: ICI -> specify immune checkpoint inhibitor Row 131: factors have been IDENTIFIED Row 138: the focus is shifting ON tumor burden 

Overall, the paper deals with a challenging aspect that does not find in literature a clear and a unique view. For this reason, an overview like this deserves to be published, but major revision of existing literature and a more comprehensive overview, even in its narrative review setting, must be done.

Response:

We are grateful for your precious comments thanks to we feel the manuscript now is greatly ameliorated.

Thank you for noticing our typos, we corrected the manuscript as noted

As correctly reported, disappearing colorectal liver metastases are a clinical dilemma primarily for the surgeon and a more dept surgical point of view has been clarified through the manuscript. 

As requested, the introduction section has been modified and the role of surgical treatment has been clarified, in particular of parenchymal sparing techniques.

Regarding the role of radiologists, as correctly reported, we did not report an important strategy which is GD-EOB-MRI. we added the paragraph in the subheading. Moreover, as requested we stressed the role of radiologists in the
multidisciplinary team.

Regarding surgical procedure, as requested we added a paragraph regarding parenchymal-sparing hepatectomy and the role of R1vasc and we clarified the role of CEIOUS and EOB-MRI, citing the study of Kokudo et al. in 2017.

Reviewer 2 Report

This narrative review addresses a clinically important and contemporary issue in the management of colorectal liver metastases (CRLM). Advancements in systemic therapy have led to disappearing CRLM (dCRLM). However, the management of dCRLM remains challenging and controversial because of the lack of evidence and consensus. The authors attempt to summarise the "state of the art" in the management of dCRLM in this review. 

1. The structure of the Introduction should be reviewed. There are eight separate paragraphs, some with only one sentence. The authors should consider consolidating the points into two or three paragraphs that present a logical flow. 

2. In the Introduction, the authors state that "Over 60% of patients with CRLM present objective response t chemotherapy according to RECIST 1.1 criteria and this is usually an indicator of favourable prognosis". This sentence is not adequately supported by the reference provided. 

3. The authors mention contrast-enhanced intraoperative ultrasound in the Introduction but it is not discussed elsewhere in the manuscript. Is there a role for intraoperative ultrasound in the assessment of dCRLM? 

4. Section 2 summarises systemic therapy for CRLM. I appreciate that it is challenging to summarise a vast amount of detail. However, again, I find that it is difficult to follow even for a reader that has some prior knowledge of this subject. Part of this is because of the loss of information when these trials are summarised in two or three sentences. For example, the study by Falcone et al. is a relevant study but when you summarise it in two sentences, the context is lost and it is difficult for a reader to make sense of it. 

5. The term "true" complete response is used in Section 3. Please define this. 

6. Table 1 appears quite inconsistent; some have P values, some have OR, some have both and some have none. 

7. Reference 39 in Table 1 is van Vledder et al (2010) but reference 39 under References is Tsilimigras et al (2019). It is essential that your references are rechecked and accurate. 

8. Your references in Table 2 do not correlate with your reference list. For example, Elias et al 2004 [48] in Table 2 correlates with Prevost et al. 2020 in your list of references at the end of the manuscript. Further, you have two Elias et al (2004 and 2007) in Table 2 but I cannot see any corresponding references. It is essential that your references are rechecked and accurate. 

9. Care needs to be exercised with language and expression. For example, the authors mention in Section 5 that "some of them have literally melted away". Please consider using accurate radiological or pathological terminology. 

10. The addition of images of dCRLM (US/CT/MRI/IUOS) could potentially add value in illustrating the problem. 

11. Can the authors propose an algorithm for the management of dCRLM based on current evidence?

12. Consider a direct answer to the question posed in your title in the Conclusion.

My main concern is that the references cited in the text do not correlate with the reference list and there are missing references. This substantially impacts the value of this review. As a reader with some prior knowledge of the subject matter, I feel that the manuscript lacks clarity. Further work is needed in terms of the organisation of the content and readability.  

Author Response

Reviewer #2,
This narrative review addresses a clinically important and contemporary issue in the management of colorectal liver metastases (CRLM). Advancements in systemic therapy have led to disappearing CRLM (dCRLM). However, the management of dCRLM remains challenging and controversial because of the lack of evidence and consensus. The authors attempt to summarise the "state of the art" in the management of dCRLM in this review. 
We are grateful for your precious comments thanks to we feel the manuscript now is greatly ameliorated.

The structure of the Introduction should be reviewed. There are eight separate paragraphs, some with only one sentence.
The authors should consider consolidating the points into two or three paragraphs that present a logical flow.

Paragraph has been consolidated as requested

In the Introduction, the authors state that "Over 60% of patients with CRLM present objective response t chemotherapy according to RECIST 1.1 criteria and this is usually an indicator of favourable prognosis". This sentence is not adequately supported by the reference provided.

The sentence has been revised with an appropriate reference to support the evidence reported.

The authors mention contrast-enhanced intraoperative ultrasound in the Introduction but it is not discussed elsewhere in the manuscript. Is there a role for intraoperative ultrasound in the assessment of dCRLM? 

CEIOUS is a valuable technique that provides useful information to decide intraoperative strategy. In fact when used in association with EOB-MRI could detect the clinical relevant DLM with viable tumor and thus at risk of local recurrence. This aspect has been clarified in-text.

Section 2 summarises systemic therapy for CRLM. I appreciate that it is challenging to summarise a vast amount of detail. However, again, I find that it is difficult to follow even for a reader that has some prior knowledge of this
subject. Part of this is because of the loss of information when these trials are summarised in two or three sentences. For example, the study by Falcone et al. is a relevant study but when you summarise it in two sentences, the context is lost
and it is difficult for a reader to make sense of it. 

Regarding systemic medical therapy, we are aware that a vast amount of detail could be challenging. As suggested we have clarified Falcone study.

The term "true" complete response is used in Section 3. Please define this. 

Thanks for your comment, the term "true" complete response was misleading and not proper, therefore we eliminated it.

Table 1 appears quite inconsistent; some have P values, some have OR, some have both and some have none. 

Many thanks for your comment, unfortunately some data are lacking, therefore we reported OR or p-value just when
available in the study,

Reference 39 in Table 1 is van Vledder et al (2010) but reference 39 under References is Tsilimigras et al (2019). It is essential that your references are rechecked and accurate. Your references in Table 2 do not correlate with your
reference list. For example, Elias et al 2004 [48] in Table 2 correlates with Prevost et al. 2020 in your list of references at the end of the manuscript. Further, you have two Elias et al (2004 and 2007) in Table 2 but I cannot see any 
corresponding references. It is essential that your references are rechecked and accurate. 

Many thanks for noticing this typos, we have manually rechecked all reference that now are correctly numbered.

Care needs to be exercised with language and expression. For example, the authors mention in Section 5 that "some of them have literally melted away". Please consider using accurate radiological or pathological terminology. 

We have revised English language of the complete manuscript, and corrected the terminology when not proper.

Can the authors propose an algorithm for the management of dCRLM based on current evidence?

An algorithm for the management of DLM based on current evidence has been provided

Consider a direct answer to the question posed in your title in the Conclusion.

Many thanks for the suggestions, this has been done.

Round 2

Reviewer 2 Report

This revised manuscript is an improvement from the initial submission. Table 1 and 2 are useful summaries of the current evidence on disappearing colorectal liver metastases. 

Does the review answer the question in the title? What do you mean by a 'ghostbuster'? The authors should consider addressing this in the manuscript/Conclusion. 

There is much uncertainty in the management of CLM. The review has addressed various components of these uncertainties but the flow is not necessarily clear. 

This sentence is unclear unclear 'Although systemic chemotherapy remains the cornerstone of is gaining popularity 37 treatment for metastatic and potentially resectable disease, surgical treatment is the only 38 definitivetreatment'. How is systemic chemotherapy the cornerstone when surgical treatment is definitive?

Are there currently proposed treatment algorithms or consensus and how does this differ from the proposed algorithm (Figure 1)? Are each of the steps in this algorithm supported by the evidence presented in the text or not conflicting with the evidence presented? For example, what is the evidence to support the watch and wait approach in the algorithm when 'microscopic residual disease or early recurrence in situ up to more than 80%'? The watch and wait approach has an arrow pointing to surgical treatment. When should the treatment strategy change?

Be cognisant about referring to surgeons as 'He' (page 6, paragraph 2, line 235). 

There are typographic errors. e.g. in Figure 1, 'multidisciplinary' is spelled as 'multidisciplinar'. The authors should thoroughly check the manuscript for typographic and grammatical errors.  

Overall, I think this review could use further revision to provide a clear and useful review of this challenging topic. 

Author Response

Dear reviewer,

Thank you again for your revision, we hope that with our further improvement the manuscript will fulfill the criteria for the Healthcare journal.

Please find attached a point-by-point response of the issues raised and our revised manuscript.

We thank you for reconsidering our paper for publication in Healthcare.

  • Does the review answer the question in the title? What do you mean by a 'ghostbuster'? The authors should consider addressing this in the manuscript/Conclusion.

Dear Reviewer, we changed the conclusion to address the request.

  • There is much uncertainty in the management of CLM. The review has addressed various components of these uncertainties but the flow is not necessarily clear. This sentence is unclear unclear 'Although systemic chemotherapy remains the cornerstone of is gaining popularity 37 treatment for metastatic and potentially resectable disease, surgical treatment is the only 38 definitive treatment'. How is systemic chemotherapy the cornerstone when surgical treatment is definitive?

Dear reviewer as requested we clarified the sentence.

  • Are there currently proposed treatment algorithms or consensus and how does this differ from the proposed algorithm (Figure 1)? Are each of the steps in this algorithm supported by the evidence presented in the text or not conflicting with the evidence presented? For example, what is the evidence to support the watch and wait approach in the algorithm when 'microscopic residual disease or early recurrence in situ up to more than 80%'? The watch and wait approach has an arrow pointing to surgical treatment. When should the treatment strategy change?

Dear reviewer, as requested figure 1 has been totally revised according to the current literature.

  • Be cognisant about referring to surgeons as 'He' (page 6, paragraph 2, line 235). There are typographic errors. e.g. in Figure 1, 'multidisciplinary' is spelled as 'multidisciplinar'. The authors should thoroughly check the manuscript for typographic and grammatical errors. 

Dear reviewer, due to the poor work provided from the previous native speaker, we sent our work to another native speaker for corrections. In the word file you will find the manuscript with the changes and in the pdf file the plain text.

Overall, I think this review could use further revision to provide a clear and useful review of this challenging topic.

We are grateful for your precious comments thanks to we feel the manuscript now is greatly ameliorated.